# Analysing Large Inconsistent Knowledge Graphs using Anti-Patterns

Thomas de Groot, Joe Raad, and Stefan Schlobach

Department of Computer Science, Vrije Universiteit Amsterdam, The Netherlands
{t.j.a.de.groot | j.raad | k.s.schlobach}@vu.nl

**Abstract.** A number of Knowledge Graphs (KGs) on the Web of Data contain contradicting statements, and therefore are logically inconsistent. This makes reasoning limited and the knowledge formally useless. Understanding how these contradictions are formed, how often they occur, and how they vary between different KGs is essential for fixing such contradictions, or developing better tools that handle inconsistent KGs. Methods exist to explain a single contradiction, by finding the minimal set of axioms sufficient to produce it, a process known as justification retrieval. In large KGs, these justifications can be frequent and might redundantly refer to the same type of modelling mistake. Furthermore, these justifications are –by definition– domain dependent, and hence difficult to interpret or compare. This paper uses the notion of anti-pattern for generalising these justifications, and presents an approach for detecting almost all anti-patterns from any inconsistent KG. Experiments on KGs of over 28 billion triples show the scalability of this approach, and the benefits of anti-patterns for analysing and comparing logical errors between different KGs.

**Keywords:** linked open data, reasoning, inconsistency

## 1 Introduction

Through the combination of web technologies and a judicious choice of formal expressivity (description logics which are based on decidable 2-variable fragments of first order logic), it has become possible to construct and reason over Knowledge Graphs (KGs) of sizes that were not imaginable only few years ago. Nowadays, KGs of billions of statements are routinely deployed by researchers from various fields and companies. Since most of the large KGs are traditionally built over a longer period of time, by different collaborators, these KGs are highly prone for containing logically contradicting statements. As a consequence, reasoning over these KGs becomes limited and the knowledge formally useless.

Typically, once these contradicting statements in a KG are retrieved, they are either logically explained [22] and repaired [20], or ignored via non-standard reasoning [13]. This work falls in the first category of approaches where the focus is to find and explain what has been stated in the KG that causes the inconsistency. Understanding how these contradictions are formed and how often

they might occur is essential for fixing and avoiding such contradictions. At least, it is a necessary step for developing better tools that can handle inconsistent KGs. For explaining contradictions, the notion of *justification*, which is a minimal subset of the KG that is sufficient for the contradiction to hold, plays a key role [12].

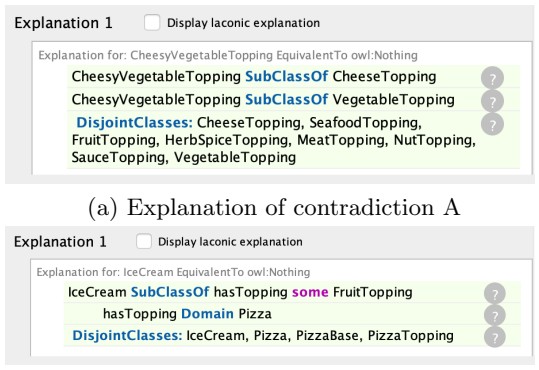

(a) Explanation of contradiction A

(b) Explanation of contradiction B

Fig. 1: Screenshot of the ontology editor Protégé showing the explanations of two contradictions found in the Pizza ontology

*Example 1.* In the renowned *Pizza ontology*[1] that serves as a tutorial for OWL and the ontology editor Protégé, we can find two contradictions that were asserted by its developers on purpose. The first contradiction (A) demonstrates the unsatisfiable class *CheesyVegetableTopping*, that has two disjoint parents *Cheese-Topping* and *VegetableTopping*. The second contradiction (B) demonstrates a common mistake made with setting a property's domain, where the class *Pizza* is asserted as the domain of the property *hasTopping*. This statement means that the reasoner can infer that all individuals using the *hasTopping* property must be of type *Pizza*. On the other hand, we find in the same ontology a property restriction on the class *IceCream*, stating that all members of this class must use the *hasTopping* property. However, since it is also specified that the classes *Pizza* and *IceCream* are disjoint, now enforcing an unsatisfiable class to have a member leads to an inconsistency in the ontology. As presented in Figure 1, justifications serve to explain such contradictions, by showing the minimal set of axioms from the ontology that causes the contradiction to hold.

Although justifications provide a good basis for debugging data quality and modelling issues in the KG, their specificity in explaining the contradictions increases in some cases the complexity for analysing and dealing with these

---

[1] https://protege.stanford.edu/ontologies/pizza/pizza.owl

detected contradictions. Particularly in large KGs, these complexities are amplified and encountered in different dimensions. Firstly, existing methods to retrieve entailment justifications do not *scale* to KGs with billions of triples. Secondly, these retrieved contradictions with their justifications can be too *frequent* to manually analyse and understand the modelling mistakes made by the ontology designer. This is especially inconvenient when a significant number of these retrieved justifications actually refer to the same type of mistake, but instantiated in different parts of the KG (e.g. similar misuse of the domain and range properties in multiple cases). Thirdly, since justifications represent a subset of the KG, they are by definition *domain dependent*, and requires some domain knowledge for understanding the contradiction. This fact is obviously more limiting in complex domains, such as medical KGs, as opposed to the Pizza ontology example above. These various challenges in finding and understanding justifications in their traditional form, poses the following research questions:

**Q1:** Can we define a more general explanation for contradictions, that categorises the most common mistakes in a KG, independently from its domain?
**Q2:** Can we retrieve these generalised explanations from any KG, independently from its size?
**Q3:** How can these generalised explanations help analysing and comparing certain characteristics between the most commonly used KGs in the Web?

This paper introduces a method for extracting and generalising justifications from any inconsistent KG. We call these generalised justifications *anti-patterns*, as they can be seen as common mistakes produced either in the modelling or population phases, or possibly stemming from erroneous data linkage. We have developed an open-source tool that can retrieve these anti-patterns from any (inconsistent) KG. We test the scalability and the completeness of the approach on several KGs from the Web, including *LOD-a-lot*, *DBpedia*, *YAGO*, *Linked Open Vocabularies*, and the *Pizza* ontology, with a combined size of around 30 billion triples. Despite deploying a number of heuristics to ensure scalability, our experiments show that our method can still detect a large number of anti-patterns in a KG, in reasonable runtime and computation capacity. Finally, we publish these detected anti-patterns in an online catalogue encoded as SPARQL queries, and show how these anti-patterns can be put to use for analysing and comparing certain characteristics of these inconsistent KGs.

The rest of the paper is structured as follows. Section 2 presents related works. Section 3 presents the preliminaries and notation. Section 4 introduces our notion of anti-patterns and describes our approach for detecting them. Section 5 presents the evaluation of the approach. Section 6 presents inconsistency analyses conducted on several large inconsistent KGs. Section 7 concludes the paper.

## 2    Related Work

Dealing with inconsistent knowledge bases is an old problem, and solutions have been proposed as early as 1958 by Stephen Toulmin [24], where reasoning over

consistent subbases was proposed. Explaining why a knowledge base entails a certain logical error (or entailment in general) has taken up traction in the last decade, leading to the most prevalent form of explanation in OWL knowledge bases called justification: minimal subsets of the graphs preserving entailments [15, 22]. A number of approaches, described in [3], aims at supporting users' understanding of *single* justifications for single entailments. Such approaches focus on reducing the axioms in justifications to their relevant parts and remove superfluous information [12, 14], providing intermediate proof steps to explain subsets of justifications [11], or attempting to improve understandability of an explanation by abstracting from the logical formalism to natural language [17].

In this work, we focus on a complementary part of the problem, where the goal is to facilitate the understandability of *multiple* justifications, of logically incorrect entailments, that share the same structure. For this, we rely on the notion of *anti-pattern* that represents a generalisation of such justifications. The term 'anti-pattern' appears in the work of [21], where the authors manually classify a set of patterns that are commonly used by domain experts in their DL formalisations, and that normally result in inconsistencies. In addition anti-patterns are also studied in [19], where the authors use a combination of reasoning and clustering methods for extracting common patterns in the detected justifications. However, this approach cannot be applied to any inconsistent KG, as it requires the KG to be mapped to the foundational ontology DOLCE-Zero.

Moreover, the notion of justifications with the same structure has been previously investigated but not formalised by [16], while the more complex notion of justification template has been introduced by [3], where several equivalence relations over justifications has been explored. In comparison with the mentioned works, this is the first work that relies on a simpler notion of justifications' generalisation, for the goal of analysing common logical errors in KGs, at the scale of the Web. Our method for detecting such anti-patterns reuses part of the work of [18], where the authors propose the efficient algorithm for path finding that we use in our subgraph generation. Our work can also be compared to an earlier large scale justification retrieval approach that uses MapReduce [26] but was mainly evaluated over synthetic data sets, and recent analyses [4, 7, 8], that aim at studying general characteristics of large KGs in the Web.

## 3   Background

In this section, we give the preliminary background and introduce the notation.

We consider a vocabulary of two disjoint sets of symbols[2]: $L$ for literals and $I$ for IRIs (Internationalised Resource Identifiers). The elements of $T = L \cup I$ are called RDF terms, and those of $I \times I \times T$ are called RDF triples. An RDF knowledge graph $\mathcal{G}$ is a set of RDF triples.

Let $V$ be a set of variable symbols disjoint from $T$, a triple pattern is a tuple $t \in (I \cup V) \times (I \cup V) \times (T \cup V)$, representing an RDF triple with some positions

---

[2] We do not consider blank nodes in this work.

replaced by variables. Any finite set of triple patterns is a basic graph pattern (BGP) $P$, and forms the basis of SPARQL for answering queries (matching a BGP to a subgraph of $\mathcal{G}$ by substituting variables with RDF terms).

We use $var(t)$ and $var(P)$ to denote the set of variables occurring in a triple pattern $t$ and a BGP $P$, respectively[3]. A substitution $\mu$ is a total mapping from $V$ to $T$ i.e., $\mu\colon V \to T$. The domain of $\mu$, denoted by $dom(\mu)$, is the subset of $V$ where $\mu$ is defined. Given a triple pattern $t$ and a mapping $\mu$ such that $dom(\mu) := var(t)$, $\mu(t)$ is the triple obtained by replacing the variables in $t$ according to $\mu$. Similarly, given a BGP $P$ and a substitution $\mu$ such that $dom(\mu) := var(P)$, $\mu(P)$ is the set of triples obtained by replacing the variables in the triples of $P$ with respect to $\mu$.

Finally, we use the standard notions of *entailment, satisfiability* and *consistency* for RDF(S) and OWL [2, 10]. Most importantly, *an inconsistent KG* is a graph for which no *model* exists, i.e. a formal interpretation that satisfies all the triples in the graph given the semantics of the used vocabularies. Let $E$ be some entailment relation, and $e$ a triple such that $\mathcal{G} \models e$. A subgraph $J(e)$ of $\mathcal{G}$ is called a justification for the inferred triple $e$, if $J(e) \models e$, and $\nexists J'$ s.t. $J' \subset J(e)$ and $J' \models e$ (i.e. a justification is a minimal subset of the knowledge graph that is responsible for the inferred triple). When $e$ is involved in a contradiction, then naturally, its justification $J(e)$ will play a key role in explaining the contradiction.

## 4  Defining and Detecting Anti-Patterns

In this section, we introduce the notion of anti-patterns, and describe our approach for retrieving anti-patterns from any inconsistent KG.

### 4.1  Anti-patterns

As previously defined, a justification is a minimal description of a single entailment (or contradiction), which can be represented as an instantiated BGP. If $\mathcal{G}$ is consistent w.r.t. some entailment relation $E$, one could call a BGP $P$ a pattern for $\mathcal{G}$ if there is a substitution $\mu(P)$ such that $\mathcal{G} \models \mu(P)$. In other words, if there is a variable assignment such that all instantiated graph patterns are entailed by the knowledge graph.

Suppose now that the knowledge graph $\mathcal{G}$ is inconsistent w.r.t $E$. In that case, trivially not every BGP would be a pattern. Therefore we define, the more interesting syntactic notion of a BGP as an anti-pattern for $\mathcal{G}$ w.r.t $E$, as a minimal set of triple patterns that can be instantiated into an inconsistent subset of $\mathcal{G}$. We define the notion of *anti-patterns* as follows.

**Definition 1 (Anti-Pattern).** *$P$ is an anti-pattern of a Knowledge Graph $\mathcal{G}$ if there is a substitution $\mu(P)$ s.t. $\mu(P) \subseteq \mathcal{G}$, and $\mu(P)$ is minimally inconsistent w.r.t some entailment relation $E$, i.e. $\nexists P'$ s.t. $P' \subset P$ and $\mu(P')$ is inconsistent.*

---

[3] Therefore, $var(t) \subseteq var(P)$ whenever $t \in P$.

For an anti-pattern $P$, we denote by $|P|$ the size of this anti-pattern, defined as the number of triple patterns in $P$. We define also the notion of *support* $sup(P)$ for an anti-pattern, as the number of substitutions $\mu(P)$. Intuitively, $\mu(P)$ refers to a particular justification in $\mathcal{G}$, and the support refers to the number of justifications occurring in the KG for a certain anti-pattern.

In order to transform a justification into an anti-pattern, we replace the elements in the subject and object position of the BGP with variables. In order to prevent breaking the contradiction, elements appearing in the predicate position of a justification are not replaced in the anti-pattern, with the exception of one case: elements appearing in the predicate position and also appearing in the subject or object position of the same justification.

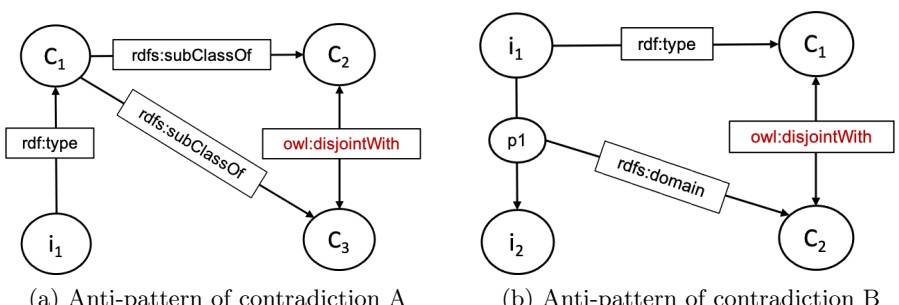

(a) Anti-pattern of contradiction A        (b) Anti-pattern of contradiction B

Fig. 2: Graphical representation of the anti-patterns of the two contradictions found in the Pizza ontology of Example 1, with circles representing the variables $V$ and rectangles representing the RDF terms $T$.

Going back to Example 1, we presented two explanations of contradictions found in the Pizza ontology. Contradiction A shows an inconsistency in the ontology in which the unsatisfiable class *CheesyVegetableTopping*, that is a subclass of the two disjoint classes *CheeseTopping* and *VegetableTopping*, is instantiated. While this example refers to a specific case of a contradiction entailed from the description of these three classes, it also refers to a common type of modelling or linking mistake that can be present in another (part of the) ontology. For instance, using the same principle, the modeller could have also created the class *FruitVegetableTopping* as subclass of the two disjoint classes *FruitTopping* and *VegetableTopping*. This formalisation of certain types of mistakes is what we refer to as anti-patterns. Figure 2 presents the two anti-patterns generalising the justifications of contradictions A and B. For instance in the anti-pattern of contradiction A, the three classes *CheesyVegetableTopping*, *CheeseTopping* and *VegetableTopping* are replaced with the variables $C_1$, $C_2$ and $C_3$, respectively. In this anti-pattern, replacing the predicate *owl:disjointWith* with a variable $p_1$ would break the contradiction, since $p_1$ could potentially be matched in the KG with another predicate such as *rdfs:subClassOf*. On the other hand, we can see in the anti-pattern of contradiction B that the predicate *hasTopping* is replaced

with the variable $p_1$, since it also appears in the subject position of the triple ⟨*hasTopping*, *rdfs:domain*, *Pizza*⟩. This allows the same anti-pattern to generalise other justifications in the KG, that follow the same pattern but involve a different property than *hasTopping*.

### 4.2   Approach

This section describes our approach for finding anti-patterns from any inconsistent KG. Finding anti-patterns from a KG would mainly consist of two steps: retrieving justifications of contradictions, and then generalising these detected justifications into anti-patterns. Such approach is expected to deal with multiple dimensions of complexity, mainly:

**Knowledge Graphs can be too large to query.** Now that KGs with billions of triples have become the norm rather than the exception, such approach must have a low hardware footprint, and must not assume that every KG will always be small enough to fit in memory or to be queried in traditional triple stores.

**Justification retrieval algorithms do not scale.** Finding all contradictions, and computing their justifications is a computationally expensive process, as it typically requires loading the full KG into memory. Therefore, when dealing with KGs of billions of triples, existing justification retrieval methods and tools do not scale [5].

Theoretically, guaranteeing the retrieval of *all* anti-patterns given any inconsistent KG requires firstly finding *all* contradictions with their justifications, and then generalising these justifications into anti-patterns. In practice, and as a way to tackle the above listed challenges, our approach introduces a number of heuristics in various steps of the approach. These heuristics emphasises the scalability of the approach, opposed to guaranteeing its completeness regarding the detection of all anti-patterns. Mainly, an initial step is introduced in the pipeline that consists of splitting the original KG into smaller and overlapping subgraphs. Depending on the splitting strategy, this step can impact the number of retrieved justifications, which in its turn can potentially impact the number of the retrieved anti-patterns. In the following, we describe the mains steps of our approach consisting of (1) splitting the KG, (2) retrieving the contradictions' justifications, and (3) generalising these justifications. Figure 3 summarises these three steps.

**1. Splitting the KGs.** Due to the large size of most recent KGs, running a justification retrieval algorithm over the complete KG to retrieve all contradictions is impractical. To speed up this process or even make it feasible for some larger KGs, we split the KG into smaller subgraphs. Each subgraph is generated by extending a root node as a starting point, that is retrieved by taking a distinct RDF term that appears in the subject position of at least one triple. As a result, the number of generated subgraphs is always equal to the number of distinct

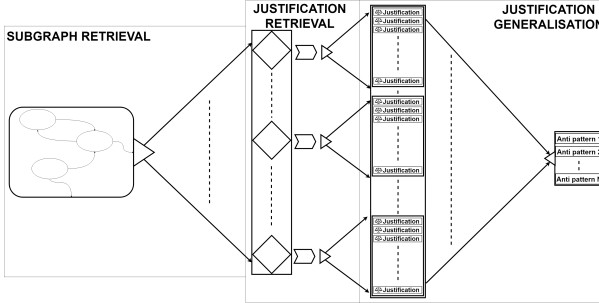

Fig. 3: Diagram that shows the pipeline used to extract subgraphs, find justifications in these subgraphs, and generalise these justifications to anti-patterns.

subjects in a KG. Using a breadth-first search, the graph is expanded by finding all the triples that have the root node as the subject, and these triples are added to the subgraph. Next, all the nodes in the object position are expanded, together with the predicates, and the graph is expanded as long as possible, or until the maximum amount of triples $G_{max}$ set by the user is reached[4]. In Section 5, we empirically estimate the optimal $G_{max}$ value for large general domain KGs, based on the trade-off between scalability of the approach and its completeness in terms of the detected anti-patterns.

**2. Justification retrieval.** Out of these newly formed subgraphs, we are only interested in the ones that are inconsistent. Therefore, we firstly check for the consistency of each of these subgraphs and discard the consistent ones. Then, for each of the inconsistent subgraphs, we run a justification retrieval algorithm to retrieve the detected inconsistencies with their justifications. For this, we use the justification retrieval algorithm in the Openllet reasoner with the OWL 2 EL profile, that walks through the graph and finds the minimal justification for each contradiction. It continues to search for justifications until no more justification can be found in the graph[5]. This step is executed for each subgraph, and all the justifications are then pushed to the final stage of the pipeline.

**3. Justification generalisation.** While most justifications are different, as each one represents a set of instantiated triple patterns, the underlying non-instantiated BGPs do not have to be. The underlying BGP forms the basis of the anti-patterns. To retrieve all anti-patterns from the detected justifications, we first generalise the justification to an anti-pattern by removing the instantiated subject and object on the nodes (when applicable, also the instantiated

---

[4] Alternatively one can use a neighbourhood radius to limit the size of a subgraph instead of $G_{max}$

[5] since justification retrieval algorithms can potentially run for a long time in the search for additional justifications in a KG, we set a runtime limit between 10 and 20 seconds based on the considered subgraph size

predicate is removed, such as the case described in Section 4.1). Justifications with the same underlying pattern are grouped together. Therefore, given a justification and its generalisation into anti-pattern, we check whether an anti-pattern with the same structure already exists. Comparing anti-patterns with different variable names consists in checking whether these anti-patterns are isomorphic. For this, we implement a version of the VF2 algorithm [6], with the addition of matching the instantiated edges of the anti-patterns (i.e. matching the predicates that do not appear in the subject or object position of the same justification). If the anti-pattern $P$ of a certain justification is matched to an existing anti-pattern, we group this justification with the other justifications generalised by this anti-pattern, and increment $sup(P)$ by 1. Otherwise, a new anti-pattern is formed as a generalisation of this justification. This algorithm continues until all justifications have been matched to their corresponding anti-patterns.

**Implementation.** The source code of our approach is publicly available online[6]. It is implemented in JAVA, and relies on a number of open-source libraries, mainly *jena*[7], *hdt-java*[8], *openllet*[9], and *owlapi*[10]. All experiments in the following sections have been performed on an Ubuntu server, 8 CPU Intel 2.40 GHz, with 256 GB of memory.

## 5   Experiments

As a way of emphasising scalability over completeness, our approach for finding anti-patterns from any inconsistent KG implements an initial step that consists of splitting the KG into smaller and overlapping subgraphs.

In the first part of these experiments (Section 5.1), we empirically evaluate the impact of the subgraph size limit $G_{max}$ on the efficiency of the approach. Then, based on the $G_{max}$ estimated from the first experiment, we show (in Section 5.2) the scalability of our approach on some of the largest KGs publicly available on the Web.

### 5.1   Completeness Evaluation

In this section, we measure the impact of splitting the KG both on the number of detected anti-patterns, and the runtime of the approach. The goal of this experiment is to ultimately find the optimal subgraph size limit to consider in the first step of the approach. For evaluating completeness, this experiment requires datasets in which *Openllet* can retrieve (almost) all inconsistencies with their justifications. For this, we rely on the three following datasets:

---

[6] https://github.com/thomasdegroot18/kbgenerator
[7] https://jena.apache.org
[8] https://github.com/rdfhdt/hdt-java
[9] https://github.com/Galigator/openllet
[10] https://owlcs.github.io/owlapi

- **Pizza ontology:** dataset of 1,944 triples serving as a tutorial for OWL. We choose this dataset based on the fact that its contradictions and anti-patterns are known, and therefore can represent a gold standard for our approach.
- **Linked Open Vocabularies (LOV):** dataset of 888,017 triples representing a high quality catalogue of reusable vocabularies for the description of data on the Web [25]. We choose this dataset since it is small enough to retrieve almost all of its contradictions and their justifications using *Openllet*.
- **YAGO:** dataset of more than 158 million triples covering around 10 million entities derived from Wikipedia, WordNet and GeoNames [23]. We choose this dataset to observe whether the optimal size limit varies significantly between the previous datasets and this relatively larger one.

In the following experiments, we vary the subgraph size limit $G_{max}$, and observe the number of detected anti-patterns and the corresponding runtime for each of the three steps of our approach. Table 1 presents the first experiment, conducted on the only considered dataset which all of its contradictions are known and can be computed on the whole graph. These results show that splitting the Pizza dataset into smaller, but overlapping, subgraphs does not impact the coverage of the approach, as both available anti-patterns in this dataset are detected even when small subgraph size limits are considered.

Table 1: Impact of the subgraph size limit $G_{max}$ on the number of detected anti-patterns and the runtime of the approach (in seconds) for the Pizza dataset.

| $G_{max}$ | Detected Anti-patterns | Total Runtime | Step 1 Runtime | Step 2 Runtime | Step 3 Runtime | Number of Subgraphs |
|---|---|---|---|---|---|---|
| 50 | 2 | 3 | 1 | 2 | 0.01 | 335 |
| 100 | 2 | 4.3 | 1.3 | 3 | 0.01 | 186 |
| 250 | 2 | 8 | 3 | 5 | 0.05 | 77 |
| 500 | 2 | 13 | 6 | 7 | 0.04 | 38 |
| 750 | 2 | 18 | 8 | 10 | 0.08 | 25 |
| 1K | 2 | 23 | 10 | 13 | 0.08 | 19 |
| No limit | 2 | 3.2 | - | 3.1 | 0.1 | - |

Table 2 presents the results of the same experiment conducted on the LOV and YAGO datasets. We adapt the considered $G_{max}$ to the size of these datasets. These results show that in both datasets, choosing a subgraph size limit of *5,000 triples* provides the optimal trade-off between the runtime of the approach and the number of detected anti-patterns for both these KGs. Moreover, and similarly to the previous experiment, we observe that the justification retrieval step (i.e. Step 2) is the most time consuming step, accounting in some cases up to 94% of the total runtime. Finally, and as it was not possible to run *Openllet* on graphs larger than 100K triples, this experiment does not guarantee for both datasets that all possible contradictions and anti-patterns can be detected when splitting the graph. Moreover, it does not guarantee that the optimal subgraph size limit

Table 2: Impact of the subgraph size limit $G_{max}$ on the number of detected anti-patterns and the runtime of the approach (in seconds) for LOV and YAGO.

| | $G_{max}$ | Detected Anti-patterns | Total Runtime | Step 1 Runtime | Step 2 Runtime | Step 3 Runtime | Number of Subgraphs |
|---|---|---|---|---|---|---|---|
| **L** | 500 | 0 | 1,783 | 216 | 1,566 | 2 | 101,673 |
| **O** | 1K | 2 | 3,505 | 429 | 3,073 | 3 | 50,960 |
| **V** | 5K | 39 | 4,525 | 668 | 3,829 | 28 | 10,218 |
| | 10K | 39 | 5,106 | 739 | 4,349 | 18 | 5,109 |
| | 25K | 39 | 5,347 | 835 | 4,493 | 18 | 2,041 |
| | 50K | 39 | 5,497 | 858 | 4,615 | 24 | 1,014 |
| | 100K | 39 | 5,758 | 946 | 4,792 | 20 | 507 |
| **Y** | 500 | 0 | 3,403 | 649 | 2,753 | 1 | 18,203,648 |
| **A** | 1K | 0 | 39,41 | 1,223 | 2,717 | 1 | 9,123,936 |
| **G** | 5K | 135 | 14,342 | 2,125 | 12,004 | 214 | 1,829,442 |
| **O** | 10K | 135 | 18,283 | 2,265 | 15,739 | 279 | 914,721 |
| | 25K | 135 | 19,174 | 2,938 | 16,013 | 223 | 365,422 |
| | 50K | 135 | 34,177 | 3,289 | 30,684 | 204 | 181,547 |
| | 100K | 135 | 68,264 | 3,976 | 64,081 | 206 | 90,773 |

for these two datasets can be generalised to other datasets. Therefore, we only consider the *5,000 triples* limit as an *estimation* for an optimal $G_{max}$ when splitting large general domain KGs, for the goal of detecting anti-patterns in any inconsistent dataset.

## 5.2  Scalability Evaluation

In the second part of these experiments, we evaluate the scalability of our approach on some of the largest KGs publicly available on the Web. In addition to the YAGO dataset, we choose the two following datasets:

- **LOD-a-lot:** dataset of over 28 billion triples based on the graph merge of 650K datasets from the LOD Laundromat crawl in 2015 [9].
- **DBpedia (English):** dataset of over 1 billion triples covering 4.58 million entities extracted from Wikipedia [1].

Based on the results of the previous experiment, we set the value of $G_{max}$ to 5,000 triples and run our approach for each of these three datasets. Table 3 shows that finding most anti-patterns from some of the largest KGs is feasible, but computationally expensive. Specifically, detecting 135 and 13 different anti-patterns in *YAGO* and *DBpedia* takes approximately 4 and 13 hours, respectively. Moreover, detecting 222 different anti-patterns from the *LOD-a-lot* takes almost a full week. This long runtime is mostly due to our naive implementation, as the most costly step of retrieving justifications could be parallelised, instead of sequentially retrieving these justifications for each subgraph.

Table 3: Results of detecting anti-patterns from three of the largest KGs in the Web: LOD-a-lot, DBpedia and YAGO.

|  | **LOD-a-lot** | **DBpedia** | **YAGO** |
|---|---|---|---|
| number of triples | 28,362,198,927 | 1,040,358,853 | 158,991,568 |
| number of distinct namespaces | 9,619 | 20 | 11 |
| number of distinct anti-patterns | 222 | 13 | 135 |
| largest anti-pattern size | 19 | 12 | 16 |
| *runtime (in hours)* | *157.56* | *13.01* | *3.98* |

## 6   KG Inconsistency Analysis

In the previous section, we showed that it is feasible to detect anti-patterns from some of the largest KGs in the Web, when the KG is split into overlapping subgraphs with a maximum size of 5,000 triples. In this section, we further analyse these retrieved anti-patterns and compare the detected logical errors between these three KGs.

### 6.1   What is the most common size of anti-patterns?

We already saw from Table 3 that the largest anti-patterns in the *LOD-a-lot*, *DBpedia*, and *YAGO* contain respectively 19, 12, and 16 edges. Looking at their size distribution, Figure 4 shows that the most common size of an anti-pattern $|P|$ ranges between *11* and *14* triple patterns for *LOD-a-lot* and *YAGO*, and between *6* and *11* triple patterns for *DBpedia*. This result, in addition to a manual verification of some of these anti-patterns, shows that most inconsistencies in *DBpedia* stem from direct instantiations of unsatisfiable classes, while the ones in *LOD-a-lot* and *YAGO* require following longer transitive chains, such as *rdfs:subClassOf* chains.

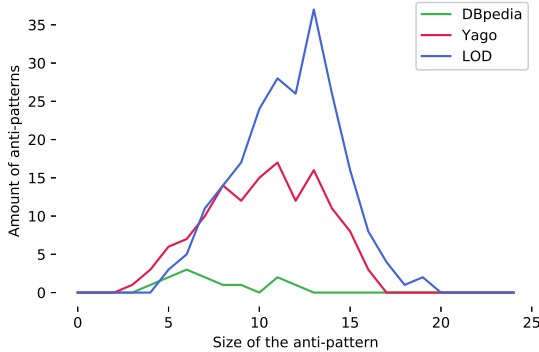

Fig. 4: Size distribution of the anti-patterns in these three KGs.

### 6.2  What are the most common types of anti-patterns found in these KGs?

Anti-patterns represent a generalised notion of justifications that describe common mistakes in a KG. In our analysis of the detected anti-patterns in these three KGs, we found that a number of the different anti-patterns refer to an even more general type of mistakes, and can be further grouped together. This general type of anti-patterns consists of anti-patterns with the same structure of nodes and edges, but with different size. Based on this principle, we can distinguish between three general types of anti-patterns found in these investigated KGs: *kite graphs*, *cycle graphs*, and *domain or range-based graphs*. Figure 5 presents a sample of detected anti-patterns referring to these three general types, and Table 4 presents their distribution in the three investigated KGs. It shows that kite graphs anti-patterns are the most common in the *LOD-a-lot* and *YAGO*, whilst cycle graph anti-patterns are the most common in *DBpedia*. All detected variants of these three general type of anti-patterns can be explored online[11].

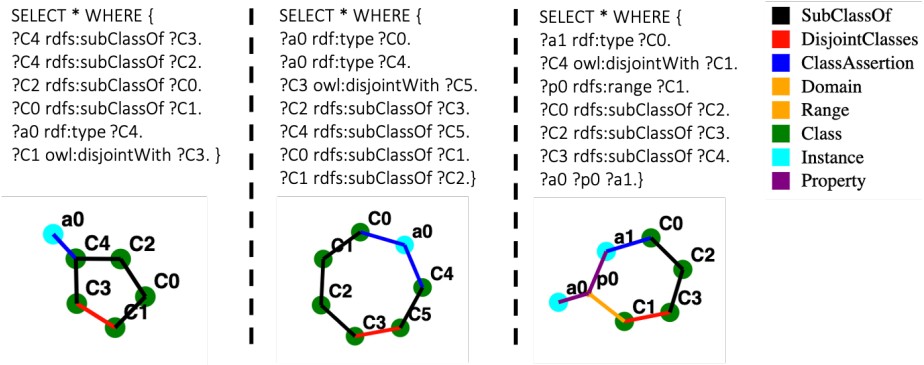

Fig. 5: Sample of three anti-patterns, referring to three general types: kite graph (left), cycle graph (middle), and domain or range-based graph (right).

Table 4: General types of anti-patterns found in these three KGs.

| Type of Anti-patterns | LOD-a-lot | DBpedia | YAGO |
|---|---|---|---|
| Kite graphs | 156 | 1 | 108 |
| Cycle graphs | 54 | 12 | 11 |
| Domain or Range-based graphs | 12 | 0 | 16 |

---

[11] https://thomasdegroot18.github.io/kbgenerator/Webpages/statisticsOverview.html

### 6.3    What is the benefit in practice of generalising justifications into anti-patterns?

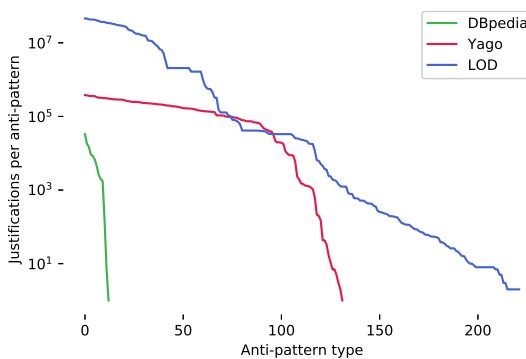

Fig. 6: Distribution of anti-pattern support.

In addition to the fact that justifications are domain-dependent and possibly complex, understanding and analysing justifications of contradictions can also be impractical due to their redundancy and frequency. This is particularly true in the three large investigated KGs, as we can see in Figure 6. This plot presents the distribution of the anti-pattern support $sup(P)$ in these three KGs. It shows that the detected anti-patterns make the millions of retrieved justifications in the *LOD-a-lot*, *DBpedia*, and *YAGO* more manageable, by generalising them into *222*, *13*, and *135* anti-patterns, respectively. Specifically, Table 5 shows that on average each anti-pattern generalises around 5M, 7.7K, and 133K justifications in the *LOD-a-lot*, *DBpedia*, and *YAGO*, respectively. It also shows that a single anti-pattern in the *LOD-a-lot* generalises more than 45M retrieved justifications. Interestingly, the *LOD-a-lot* –a dataset that represents the largest publicly available crawl of the LOD Cloud to date– contains over a billion justifications, while *DBpedia* and *YAGO* –two of the most popular available RDF datasets– contain around 100K and 18M justifications, respectively. Thus, indicating that the quality of *DBpedia* (0.0009%), estimated by the number of detected justifications per triple in $\mathcal{G}$, is significantly higher in comparison with *LOD-a-lot* (3.9%) and *YAGO* (11.3%).

## 7    Conclusion

In this work, we introduced anti-patterns as minimal sets of (possibly) uninstantiated basic triple patterns that match inconsistent subgraphs in a KG. We can use anti-patterns to locate, generalise, and analyse types of contradictions. Retrieving contradictions from a KG and finding the extent to which a KG is inconsistent can now be formulated as a simple SPARQL query using anti-patterns

Table 5: Impact of generalising justifications to anti-patterns

| $sup(P)$ | LOD-a-lot | DBpedia | YAGO |
|---|---|---|---|
| *Minimum* | 2 | 1 | 1 |
| *Maximum* | 45,935,769 | 32,997 | 379,546 |
| *Average* | 4,988,176.9 | 7,796.07 | 133,998.31 |
| *Median* | 23,126 | 4,469 | 106,698 |
| *Total* | 1,107,375,273 | 101,349 | 18,089,773 |
| *Total per triple* | 3.9% | 0.009% | 11.3% |

as BGPs. Our second contribution is a tool that can extract a large number of anti-patterns from any inconsistent KG. For evaluating our approach, we showed on relatively small KGs that our approach can detect in practice all anti-patterns despite splitting the KG, and showed on KGs of billions of triples that our approach can be applied at the scale of the Web. Specifically, we showed on the *LOD-a-lot*, *DBpedia*, and *YAGO* datasets that billions of justifications can be generalised into hundreds of anti-patterns. While these findings prove the spread of billions of logically contradicting statements in the Web of Data, this work also shows that these contradictions can now be easily located in other KGs (e.g. using a *SELECT* query), and possibly repaired (e.g. using a *CONSTRUCT* query). The source code, as well as the list of detected anti-patterns from these KGs are publicly available as SPARQL queries, with their support in each dataset.

We are aiming to extend this work by (1) including additional datasets such as Wikidata and additional commonly used domain specific datasets, (2) exploiting the previously computed transitive closure of more than half a billion *owl:sameAs* links [4], 3 billion *rdf:type* statements with 4 million *rdfs:subClassOf* in the LOD-a-lot, for the goal of detecting additional types of anti-patterns, and (3) analysing the origins of these anti-patterns which consists of analysing billions of detected justifications, and in the case of the LOD-a-lot dataset also consists of obtaining the provenance of each statement from the LOD Laundromat crawl.

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
