# OpenReview forum: "Analysing Large Inconsistent Knowledge Graphs using Anti-Patterns"
_eswc-conferences.org/ESWC/2021/Conference/Research_Track — ESWC 2021 Research_

### Official Review · AnonReviewer3 · 2021-01-05
**Reasonable technical contribution, but limited wider analysis**

**Rating:** 1
**Confidence:** 3
**Impact:** 3
**Design And Technical Quality:** 4

**Review:**

This paper presents a method for efficiently discovering minimal contradictions in ontologies, and to lift these to abstract patterns. Understanding recurrent abstract patterns would presumably be helpful to understand how and why contradictions are introduced in the first place, and how they can be avoided.

The technical contribution appears overall solid, using a partitioning strategy to retrieve contradictions over subgraphs, which is shown to not result in major incompleteness.

At the same time, unfortunately, the wider analysis appears rather limited. Contradictions are a topic of great relevance to the community, and a careful analysis of how and why these occur, and what can be learned from that, would have potential for great impact. Such an analysis would require to dig deep into the found patterns, perhaps interviewing ontology engineers, or digging into the revision history how such contradictions arose. The current paper stops at the stage where patterns are mined, unfortunately missing out any discussion of examples, presents only a superficial analysis in terms of pattern size and meta-pattern clustering.
I understand that an in-depth analysis of how and why would be a major endeavor, yet as it stands, the insights beyond the presented method are very limited.

Detailed points:
 - Intro: I was a bit confused by "once contradicting statements are retrieved, they are either explained ..." and "this work falls into the category of works that retrieve and explain" - what is meant by explanation here? Aren't minimal sets of contradictions their own explanation?
 - "the knowledge becomes formally useless" - "formally", but not "practically"?
 - I was also confused by the term anti-pattern - aren't these still patterns in the pragmatic sense of repeated structures?
 - The choice of the KGs would benefit from some motivation - why isn't the arguably most popular public KG of today, Wikidata, there?
 - The history of the KGs would also be relevant for analysing the origin of contradictions - e.g., how many people worked on these (inter-person-coordination), and how long (temporal coordination)?
 - References formatting is messy

**Anonymity:**

Yes, I would like my review to remain anonymous.

**Reuse And Availability:**

4: High

**Strong Points:**

Solid technical contribution and analysis

**Subreviewer:**

I submitted this review.

**Weak Points:**

Lack of qualitative insights, work only focuses on retrieving patterns, but does not analyze their origin, nor what can be learned from them.

---

> ### Author Rebuttal · Authors · 2021-01-29
>
> We thank the reviewer for their feedback and suggestions. We will address in the final version the minor points raised by the reviewer, and in the following we answer their main concern:
>
> > Lack of qualitative insights, work only focuses on retrieving patterns, but does not analyze their origin, nor what can be learned from them.
>
> We definitely agree that an in-depth analysis of the origins of these detected anti-patterns would be beneficial to the community, and we thank the reviewer for their suggestions for conducting such an analysis. Indeed, this paper is limited to the following three contributions: (i) defining our notion of anti-patterns, (ii) introducing our approach for detecting most anti-patterns from any RDF KG independently from its domain and size, while testing the completeness and scalability of the approach, and (iii) preliminary comparison between three commonly used KGs of different sizes, regarding the detected anti-patterns' size distribution, their type, and the number of justifications they generalise. Therefore, the main focus of this paper is not intended to address specific modelling errors in the chosen KGs, but to showcase a method that can be applied as a generic Knowledge Engineering approach, independent of the specific patterns identified in this paper.
>
> We are aiming to extend this work by (1) including additional datasets such as Wikidata and additional commonly used domain specific datasets, (2) exploiting the previously computed transitive closure of more than half a billion owl:sameAs links, 3 billion rdf:type statements with 4 million rdfs:subClassOf in the LOD-a-lot, for the goal of detecting additional types of anti-patterns, and (3) analysing the origins of these anti-patterns which consists of analysing billions of detected justifications, and in the case of the LOD-a-lot dataset also consists of obtaining the provenance of each statement from the LOD Laundromat crawl (since LOD-a-lot is the merge of 650K datasets from the LOD Laundromat crawl). We will clarify these points in the conclusion of the paper.

---

> > ### Comment · AnonReviewer3 · 2021-01-31
> > **Thanks for the response; I acknowledge the technical contribution, I don't think scaling up is the way towards more insights**
> >
> > I acknowledge the response and appreciate the sincerity of the authors in clarifying what is in-scope and out-of scope. I see that the technical contribution has some value for further research that truly aims to understand and explain these inconsistencies. I thus have upgraded my score and recommend acceptance.
> >
> > At the same time, I remain skeptical of the directions hinted at in the planned extensions. I do not think that scaling this up to further and/or larger datasets, nor automatically processing provenance triples of billions of detected justififications, is the way to insights. I presume insight would come from carefully, manually looking at a handful of samples, deriving hypotheses (about origin, reason for longevity, etc.) from these, then only verifying these hypotheses at larger scale.

---

### Official Review · AnonReviewer2 · 2021-01-11
**Original solution to an important research problem**

**Rating:** 1
**Confidence:** 4
**Impact:** 4
**Design And Technical Quality:** 3

**Review:**

This paper proposes a novel approach to deal with with inconsistent large Knowledge Graphs (KGs). The authors define the notion of anti-pattern to group several types of contradictions, propose a framework to compute them, implement it, and evaluate it. The framework includes many heuristics to guarantee a good scalability of the system. The implementation is publicly available. The evaluation shows that the framework can compute anti-patterns in a reasonable time, identifies the most common types of anti-patterns, and quantifies the reduction in the number of justifications.

The paper seems technically correct and original. The research problem is important and I think that the contributions are significant.

The main limitations are that the evaluation uses a server, whereas most users would use a machine with a much less powerfull hardware, and that the impact of the heuristics in the reduction of the running time is not evaluated. It is already known that some KG algorithms (without algorithms) do not scale well, and the implemented algorithm finishes in a reasonable time, but a detailed evaluation of this fact is not performed.

Presentation is good in general, given the space limitations, but the list of heuristics used by the authors is not clear. Some of them (or maybe all of them) are mentioned at different parts of the text, but I suggest grouping all of them in a table, subsection, etc.

Furthermore, the explanation of why some predicates are not replaced (page 6) could be improved. In Example 1, it is clear that owl:disjointWith should not be replaced. However, according to the text, the exceptions are predicates that appears in a subject or object position, but in Example 1 owl:disjointWith does not seem to appear in a subject or object position.

Minor comments:

- I agree that KGs are highly prone for containing logically contradicting statements, but the claim (in the abstract) that a *significant* number of KGs contain contradicting statements should be supported with references.

- In Section 4 there are some references supporting the claims that explanations increase the complexity and that KG algorithms do not scale well. I would also use (some of) these references in the introduction (first paragraph of page 3).

- In Figure 5, top, middle and bottom should be replaced with left, middle, and right.


*********************************************

Update: Thank you very much for the rebuttal.


**Anonymity:**

Yes, I would like my review to remain anonymous.

**Reuse And Availability:**

4: High

**Strong Points:**

- The approach is implemented in a publicly available tool.

- The results are significant to reason with the Web of data in a reasonable time.

- The top 3 families of antipatterns are identified.

**Subreviewer:**

I submitted this review.

**Weak Points:**

- Evaluation uses a server rather than a common machine for most users.

- The impact of the heuristics in the running time is not evaluated.

- The complete list of heuristics is not clear.

---

> ### Author Rebuttal · Authors · 2021-01-29
>
> We thank the reviewer for their feedback and suggestions. We will address in the final version the minor points raised by the reviewer, and in the following we answer their main concerns:
>
> > Evaluation uses a server rather than a common machine for most users.
>
> We agree that a relatively standard machine could have been used for certain experiments, as the presence of a large memory and computational power is not a requirement in these experiments due to (i) our use of HDT for compressing and querying the large KGs, and (ii) our use of Opellet only on relatively small subgraphs after splitting. For instance, the largest dataset we use is the LOD-a-lot (28 billion triples), which according to its creators [1], can be deployed on a standard laptop, as it only requires 524 GB of disk space and 15.7 GB of RAM. Therefore, the use of the server was mostly due to convenience, as running these experiments on a local machine would be doable but not recommended for large datasets that require a long runtime, hence making the experiments performance results less comparable.
>
> [1] Fernández, J. D., Beek, W., Martínez-Prieto, M. A., & Arias, M. (2017, October). LOD-a-lot. In International semantic web conference (pp. 75-83). Springer, Cham.
>
> > The complete list of heuristics is not clear.
>
> In this work, we employ the following two heuristics:
>
> 1. (H1) Splitting the KG into overlapping subgraphs with a size limit, since it is not feasible to directly retrieve justifications for web-scale KGs, such as the LOD-a-lot. We empirically estimate the optimal subgraph size limit in Table 2.
>
> 2. (H2) We set a runtime limit for the justification retrieval algorithm between 10 and 20 seconds depending on each subgraph size, since justification retrieval algorithms can potentially run for a long time in the search for additional justifications.
>
> We will clarify these heuristics in the final version, by stating them together at the beginning of Section 4.2.
>
> > The impact of the heuristics in the running time is not evaluated.
>
> We agree that the impact of (H2) on the runtime was not evaluated, due to the direct relation between increasing the runtime limit of the justification retrieval algorithm and the overall runtime of the approach. Instead, we have evaluated in Table 2 the impact of the size limit in (H1) on the runtime of each step of our approach. This evaluation shows that in both datasets, choosing a subgraph size limit of 5K triples provides the optimal trade-off between the runtime of the approach and the number of detected anti-patterns for both these KGs. Moreover, we observed that the justification retrieval step (i.e. Step 2) is the most time consuming step, accounting in some cases up to 94% of the total runtime.

---

### Official Review · AnonReviewer5 · 2021-01-13
**I recommend publishing the paper.**

**Rating:** 3
**Confidence:** 4
**Impact:** 4
**Design And Technical Quality:** 5

**Review:**

This paper presents a procedure to identify anti-patterns in knowledge graphs (KGs). The proposed steps are

1. _Splitting the KG_. For each entity $e$ that is subject in a triple, a subgraph with root $e$ is generated.
2.  _Justification retrieval_. The inconsistent subgraphs are identified.
3.  _Justification generalisation_. Each particular subgraph with inconsistencies is transformed in a subgraph with variables.

These steps are evaluated to know how the number of anti-patterns and the time required to identify them evolve according to the pre-established maximum size of the subgraphs. The scalability of the method is checked using large KGs. The most common anti-patterns in different well known KGs are also identified.

According to my point of view, the paper can be published in the conference proceedings. Nevertheless, the first sentence of the abstract is not clear for me.

> Based on formal semantics, most of the KGs on the Web of Data can be put to practical use.

**Anonymity:**

Yes, I would like my review to remain anonymous.

**Reuse And Availability:**

5: Very High

**Strong Points:**

The work is very well explained, the results are interesting, and the results are reproducible. The evaluation is convincing and provides essential information about the proposed solution.

**Subreviewer:**

I submitted this review.

**Weak Points:**

The first sentence of the abstract should be replaced or better explained.

---

> ### Author Rebuttal · Authors · 2021-01-29
>
> We thank the reviewer for their very positive feedback. We will change the first sentence of the abstract in the final version.

---

### Official Review · AnonReviewer4 · 2021-01-15
**The paper presents a truly practical approach for identifying general patterns that cause the inconsistency of knowledge graph (KG), whereas the theoretical properties of the approach remain unknown.**

**Rating:** 2
**Confidence:** 5
**Impact:** 3
**Design And Technical Quality:** 4

**Review:**

The paper (1) presents to generalize concrete triples responsible for logical errors of KG (called justifications) into patterns, which then can be reused and compared across KGs.  The idea is meaningful and important for the quality of KGs from the perspective of logical soundness, which has not been very loud in the KG community where numerical ways prevail over logical ones.  Significantly, the paper (2) demonstrates the effectiveness of the approach for large KGs including LOD, DBpedia and YAGO, and a group of patterns have been detected and analyzed in detail. All these are available online, which shall surely benefit the Semantic Web and KG people.  The paper (3) is clearly written and easy to follow. Based on all these pros, I recommend Accept to the ESWC 2021.

The cons are discussed as follows.

The approach proposed in the paper is a practical one with heuristics and settings so as to limit the  search for feasibility and scalability purposes. As claimed in the paper on Page 7, "These heuristics emphasises the scalability of the approach, opposed to guaranteeing its completeness...".  While completeness can be sacrificed, the soundness cannot, especially for an approach in pursuit of logical correctness. (1)There is none soundness proof in the paper, nor any discussion about the issue. Particularly, for the operation of Splitting the RDF graph of KG into subgraphs on Pages 7-8, it's unknown whether the splitting keeps any logical equivalence, or one subgraph being logically inconsistent  can infer the inconsistency of the original graph. Papers like "Gu, et al., Reasoning and querying web-scale open data based on DL-LiteA in a divide-and-conquer way. Journal of Web Semantics, Volume 55, March 2019, Pages 122-144" can be referred to for partitioning PDF graphs with soundness and if possible, completeness.

(2) Ontologies and KGs, although may look like similarly in representation in PDF graphs, are distinctive concepts by design. Ontologies are normally constructed by domain experts and ontologists and focus on schema knowledge facilitated by logical reasoning, whereas KGs are generated mostly by automatic search and integration and focus on individual data facilitated by numerical training and prediction. The latter is generally much larger in scale and noisier in quality than the former. Logically inconsistent LOD, DBpedia and YAGO have been used in many practical tasks such as QA and link prediction, not via logical reasoning ways. Therefore it's crucial to show how logical inconsistency can jeopardize the application of KG. This is not made clear in the paper, which as a matter of fact does not seem to distinguish between ontologies and KGs. Using the pizza ontology as an example in Section1 Introduction is a bad idea and should be replaced with part of a real-world KG.  The experiment on the pizza ontology should be removed as well, unless you argue the applicability of your approach includes ontologies, which would require completeness then.   Moreover, How the traditional way (I suppose its for ontologies) fails to scale in searching for justifications  in KGs should be elaborated with complexity, tractability and runtime analysis.

A few other smaller points as listed as follows.
- page 5, last paragraph in Section 3,  justification J(e) for triple e is unique or not? And if not, how to account for this in the pattern generalization?
- page 5, Definition 1, "inconsistency wrt some entailment relation E" has not been defined previously.
-  several occurrences of "larger KGs" should be "large KGs".
- page 11, the last sentence before Section 5.2, "for the goal of detecting almost all anti-patterns", remove "almost all", as you do not know the "degree" of completeness of your approach at all.
- page 11, the last sentence "the results suggest ...", what does this conclusion mean? How "more diverse" is defined?
- page 12, Section 6 KGs inconsistency Analysis, change "KGs" to "KG".
- page 12, last sentence of Section 6.1, "This suggests ...", how is this conclusion drawn on? How can you know the content solely from the number of triples?
- page 13, Fig 5, the figure caption, "(top)" should be "(left)", and "(bottom)" should be "(right)".




**Anonymity:**

Yes, I would like my review to remain anonymous.

**Reuse And Availability:**

4: High

**Subreviewer:**

I submitted this review.

---

> ### Author Rebuttal · Authors · 2021-01-29
>
> We thank the reviewer for their positive feedback and suggestions. We will address in the final version the minor points raised by the reviewer, and in the following we answer their main concerns:
>
> > There is none soundness proof in the paper, nor any discussion about the issue. Particularly, for the operation of Splitting the RDF graph of KG into subgraphs on Pages 7-8, it's unknown whether the splitting keeps any logical equivalence, or one subgraph being logically inconsistent can infer the inconsistency of the original graph.
>
> There are two ways to interpret this question: 1) Whether soundness of the classical reasoning is guaranteed, i.e. are all entailed formulas still entailed after splitting. This is not the case, as we start from inconsistent KGs, which entails everything. Therefore any consistent subset entails fewer triples. 2) Whether anti-patterns for the subgraphs are also anti-patterns of the original KG. This follows directly from our definition of anti-patterns, but we will explicitly mention it in the paper.
>
> > Logically inconsistent LOD, DBpedia and YAGO have been used in many practical tasks such as QA and link prediction, not via logical reasoning ways. Therefore it's crucial to show how logical inconsistency can jeopardize the application of KG. This is not made clear in the paper, which as a matter of fact does not seem to distinguish between ontologies and KGs. Using the pizza ontology as an example in Section1 Introduction is a bad idea and should be replaced with part of a real-world KG.
>
>
> We agree that we did not address the difference between ontologies and KGs in this paper, as the main aim of our work was to design an approach that can detect most anti-patterns from any RDF graph, regardless of how it was constructed, the domain it covers, and its size. We also agree that we do not argue strongly enough in the paper what the damage of inconsistency is, when users ignore the logical fact. We will add (an instance of) one of the anti-patterns we detected in the experiments to the introduction, and show how this can lead to wrong results in a query. Finally, our choice for using the Pizza ontology as an example was solely motivated by the fact that it is a widely known inconsistent RDF graph that can be used to highlight the type of problems we want to address. We will search for a similar RDF graph to replace the Pizza ontology in the final version.

---

> > ### Comment · AnonReviewer4 · 2021-01-30
> > **My review comments agreed by the authors.**
> >
> > It's good to see that the authors are willing to address the problems I commented in their revised version.  Here I would emphasize again the importance of discussing the theoretical soundness of the approach, or at least acknowledging its absence in the paper. The claim like "start from inconsistent KGs" in the rebuttal is suspicious as I doubt that current DL reasoners can digest KGs sized as largely as LOD, DBpedia and YAGO. This says that you do not know at the first place whether the KG you are dealing with is logically consistent or not. Again this goes back to the necessity of theoretical analysis, e.g., the splitting as the first step of the approach can keep the logical consistency or not.

---

### Official Review · AnonReviewer1 · 2021-01-16
**Nice idea, but could be worth a lot more.**

**Rating:** 1
**Confidence:** 4
**Impact:** 3
**Design And Technical Quality:** 4

**Review:**

This paper proposes a solution for detecting antipatterns representing inconsistency/unsatisfiability in large knowledge graphs (KGs). Here, antipatterns are basic graph pattern (i.e., triple patterns with variables allowed, as used by SPARQL) that capture minimal inconsistent sets of ontology axioms in the KG where inconsistency is defined based on RDFS or OWL semantics. The authors presented both technical details of the approach and experimental evaluation justifying it.

The paper is nicely written. The problem and the technical pieces of the solution are well-presented. The strategy to deal with large KGs that cannot fit in memory, I think, is very much appropriate to the problem. Given the motivation of the detection algorithm, evaluation suitably looks at completeness and scalability issues.

Now, looking at the procedure, it seems that in order to obtain the antipatterns, we essentially have to go through the process of computing (a lot of) justifications. So detecting antipatterns is at least as difficult as computing justifications. The result of this process is a set of antipatterns that capture inconsistency in the *given* ontology/KG. The question is then what's next after getting the set of anti patterns. How can these antipatterns be used to analyze/understand the modeling mistake? Section 6.3 should have covered this more clearly.

In terms of using antipatterns in comparing different KGs, evaluation and discussion essentially only showed that each compared KG has a particular number of a particular antipattern, and how many justifications are represented by those antipatterns. It is unclear to me what is the implication for a KG having a certain number of antipatterns of a certain type. Moreover,  it is also not apparent to me if a set of antipatterns obtained from one ontology/KG could be immediately useful/reused in the context of another ontology/KG -- Reuse is usually one of the motivation for using (anti)patterns. As the process of getting the antipatterns involves a lot of computation, it would certainly worth a lot more if such antipatterns can be useful in the context of another KG.

Finally, concerning evaluation, there are two question marks. First, in the completeness evaluation, we don't know if those obtained numbers in Table 2 cover all justifications in the KGs. So, I'm not sure if calling it completeness is appropriate. Maybe adding "approximate" would make more sense? Second, scalability evaluation seems to only shows that finding antipatterns take a certain relatively high number of hours to complete for LOD-a-lot, DBPedia, and Yago. How do argue that this answers the motivating requirement of "Justification retrieval algorithms do not scale"?

Overall, I don't mind if the paper is accepted, though I certainly won't fight for it.

**Anonymity:**

Yes, I would like my review to remain anonymous.

**Reuse And Availability:**

3: Medium

**Strong Points:**

- Paper is easy to read.
- Technical solution is appropriate to the problem.

**Subreviewer:**

I submitted this review.

**Weak Points:**

- It is not clear how the obtained antipatterns can be used to analyze the modeling mistake.
- The use of antipatterns after obtaining them through an expensive process is not clearly discussed. (Reuse in other contexts).
- Scalability evaluation does not really answer the motivating requirement of "Justification retrieval algorithms do not scale"

---

> ### Author Rebuttal · Authors · 2021-01-29
>
> We thank the reviewer for their feedback and suggestions. In the following, we answer the reviewer’s main concerns:
>
> >It is not clear how the obtained antipatterns can be used to analyze the modeling mistake.
>
> Indeed, the paper does not clarify this point, as the main contribution of this paper is not intended to address specific modelling errors in the chosen KGs, but to showcase a method that can be applied as a generic Knowledge Engineering approach, independent of the specific patterns identified in this paper. However, we believe that publishing the set of all the detected anti-patterns as SPARQL queries with their support for each dataset [1], allows dataset maintainers to better find and understand the common mistakes made in each dataset, and see their impact. Most importantly, it can allow them to easily fix these present inconsistencies (e.g. using a CONSTRUCT query). Nonetheless, we agree that this point needs to be clarified in the final version.
>
> [1] https://thomasdegroot18.github.io/kbgenerator/Webpages/statisticsOverview.html
>
>
> > The use of anti-patterns after obtaining them through an expensive process is not clearly discussed. (Reuse in other contexts).
>
> We agree that this point was not made clear in the paper, although it is a very important feature of anti-patterns. Indeed anti-patterns are designed to be reused across different KGs since they are defined as basic graph patterns, with their elements in the subject and object positions being variables, and the elements in the predicate position being typically common semantic relations that are used across different KGs (e.g. owl:disjointWith, rdf:type, rdfs:domain, etc.). Obtaining a large collection of different types of common anti-patterns, facilitates locating these anti-patterns in other KGs, as it transforms the computationally expensive justification retrieval task to a basic graph pattern matching task (e.g. using SPARQL). We will clarify this point in the final version.
>
> > Scalability evaluation does not really answer the motivating requirement of "Justification retrieval algorithms do not scale".
>
> We mention that justification retrieval algorithms do not scale as these approaches typically require loading the full KG into memory. Therefore, making the retrieval of justifications for web scale datasets not possible on a standard machine, or even a standard server. This is the main motivation for the first step of our approach, consisting of splitting the KG into overlapping subgraphs, in order to make the retrieval of justifications feasible. The long runtime in the experiments is mostly due to our naive implementation, as the most costly process of retrieving justifications could be parallelised, instead of sequentially retrieving these justifications for each subgraph. We will clarify this point in the final version.

---

### Decision · Program_Chairs · 2021-02-23

**Decision:**

Accept

**Comment:**

All reviewers unanimously agree that the paper should be accepted. They recognise the merit of this work and point out to some limitations that would certainly improve the understanding of this problem. Some of such limitations concern the nature of the antipatterns, formal justifications of a some claims, and a more in-depth analysis of some results. We encourage the authors to include, as much as possible, the reviewers' suggestions in the camera-ready version of this paper.